# SafeBench: A Benchmarking Platform for Safety Evaluation of Autonomous Vehicles

*Chejian Xu[1], *Wenhao Ding[2], Weijie Lyu[1], Zuxin Liu[2],
Shuai Wang[2], Yihan He[2], Hanjiang Hu[2], Ding Zhao[2], Bo Li[1]
[1]University of Illinois at Urbana-Champaign   [2]Carnegie Mellon University
{chejian2,wlyu3,lbo}@illinois.edu, dingzhao@cmu.edu
{wenhaod,zuxinl,shuaiwa2,yihanhe,hanjianh}@andrew.cmu.edu

## Abstract

As shown by recent studies, machine intelligence-enabled systems are vulnerable to test cases resulting from either adversarial manipulation or natural distribution shifts. This has raised great concerns about deploying machine learning algorithms for real-world applications, especially in safety-critical domains such as autonomous driving (AD). On the other hand, traditional AD testing on naturalistic scenarios requires hundreds of millions of driving miles due to the high dimensionality and rareness of the safety-critical scenarios in the real world. As a result, several approaches for autonomous driving evaluation have been explored, which are usually, however, based on different simulation platforms, types of safety-critical scenarios, scenario generation algorithms, and driving route variations. Thus, despite a large amount of effort in autonomous driving testing, it is still challenging to compare and understand the effectiveness and efficiency of different testing scenario generation algorithms and testing mechanisms under similar conditions. In this paper, we aim to provide the *first* unified platform SafeBench to integrate different types of safety-critical testing scenarios, scenario generation algorithms, and other variations such as driving routes and environments. In particular, we consider 8 safety-critical testing scenarios following National Highway Traffic Safety Administration (NHTSA) and develop 4 scenario generation algorithms considering 10 variations for each scenario. Meanwhile, we implement 4 deep reinforcement learning-based AD algorithms with 4 types of input (e.g., bird's-eye view, camera) to perform fair comparisons on SafeBench. We find our generated testing scenarios are indeed more challenging and observe the trade-off between the performance of AD agents under benign and safety-critical testing scenarios. We believe our unified platform SafeBench for large-scale and effective autonomous driving testing will motivate the development of new testing scenario generation and safe AD algorithms. SafeBench is available at `https://safebench.github.io`.

## 1 Introduction

Innovations driven by recent progress in machine learning (ML) have shown human-competitive performance in sensing [1], decision-making [2], and manipulation [3]. However, several studies have shown that when such powerful ML models are exposed to adversarial attacks they can be fooled, evaded, and misled in ways that would have profound security implications: image recognition, natural language processing, and audio recognition systems have all been attacked [4, 5, 6, 7]. As ML-based models and approaches have expanded to real-world *safety-critical applications*, such as Autonomous Driving (AD), the question of safety is becoming a crux for the transition from theories

---

*Equal Contribution

to practice [8, 9], and it is vitally important to quantitatively and efficiently evaluate the robustness or safety of safety-critical applications before their massive production and deployment. As listed in the *National Artificial Intelligence Research and Development Strategic Plan* [10], developing effective evaluation methods for AI and ML is considered one of the top priorities. Failing to meet this demand will cause death, stifle innovations, and hurt our economy, among other socially responsible issues.

*Challenges.* Despite the great importance of safety evaluation for AD algorithms, it is challenging to comprehensively and quantitatively evaluate AD algorithms due to both *real-world data* and *evaluation design* challenges. First, in practice, the safety-critical driving scenarios are "rare" – can be found by driving every 30,000 miles [11], which leads to the fact that current AD testing requires driving millions of miles with large economic and environmental costs. In addition, such rarity also requires the evaluation methods to have an accelerated feature with a probabilistic convergence guarantee to avoid being over-optimistic. Previous work [12, 13] solve this problem for abstract simple models by using large deviation theories such as importance sampling (IS) and cross entropy (CE) [14]. However, these approaches are shown to have reached bottlenecks when dealing with ML algorithms with increasing complexity. In fact, recent studies [15] have shown that these classical IS/CE based approaches and tools may consistently underestimate the risk when dealing with complex systems. Moreover, such peril has been identified in different evaluation approaches [16, 17, 18, 19, 20, 21], which have already been adopted by industry [22] and test agencies [23] in the U.S. to assess the safety of AVs. Second, although several learning-based scenario generation approaches are later proposed to overcome the above challenge [24, 25, 26, 27], existing evaluation tools and platforms are usually based on their own design, such as dataset selection, safety-critical scenario definition and generation, evaluation metrics, and input types. This makes it very challenging to fairly compare different AD algorithms or interpret different evaluation results.

In this paper, we focus on designing and developing the *first* unified **robustness and safety evaluation platform for AD algorithms, SafeBench**. In particular, we design SafeBench based on the open-sourced simulation platform Carla [28]. SafeBench consists of 4 modules, including *Agent Node*, *Ego Vehicle*, *Scenario Node*, and *Evaluation Node*. Based on our platform, we systematically evaluate the AD algorithms on 2,352 generated safety-critical testing scenarios, such as *Straight Obstacle* and *Lane Changing* together with other benign scenarios. For each safety-critical *scenario*, we implement 4 scenario *generation algorithms* for comparison. In addition, for each scenario, we select 10 diverse *driving routes* to ensure the generalization of our evaluation results. We report the evaluation results based on 10 metrics, such as *collision rate*, *frequency of running red lights*, and *average percentage of route completion*. Finally, we developed 4 reinforcement learning-based AD algorithms with different perceptual capabilities on SafeBench. Specifically, we provide 4 input types, ranging from low-dimensional state representations to complicated visual inputs. Based on our comprehensive evaluation, we find that (1) there is a performance trade-off for different AD algorithms under benign and safety-critical scenarios, (2) some safety-critical scenarios have higher transferability across AD algorithms, (3) different scenario generation algorithms achieve different levels of effectiveness even when generating the same scenario, (4) different AD algorithms achieve advantages over others under different metrics. Our findings suggest that testing AD algorithms on high-quality safety-critical scenarios is necessary and can largely improve testing efficiency, and we should consider a combination of testing scenarios and generation algorithms for effective testing.

*Contributions.* In this work, we aim to provide the first **unified** evaluation platform for different AD algorithms by generating diverse safety-critical scenarios with different generation algorithms and evaluation metrics. Our evaluation platform SafeBench includes the following properties.

- **Unified benchmarking platform with modularized design**. Our evaluation platform consists of 4 modules, including *Ego vehicle*, *Agent*, *Scenario*, and *Evaluation*. It is also flexible to replace, add, or delete modules for future functionalities and evaluations.

- **Comprehensive coverage of safety-critical scenario generation**. In SafeBench, we have integrated 2,352 testing scenarios, which have provided comprehensive coverage of known safety-critical scenarios in the real world, and it is flexible to add more testing scenarios by applying generation methods on new template scenarios.

- **Comprehensive coverage of scenario generation algorithms**. For each testing safety-critical scenario, we developed 4 generation algorithms, so that we are able to not only evaluate AD safety on the scenario level, but also on the generation algorithm level.

Table 1: Comparison of Evaluation Platforms

| Simulator | Safety-critical Scenarios | | Realistic Perception | Customized Scenario | Backend | Baselines |
| | Adversary-based | Knowledge-based | | | | |
|---|---|---|---|---|---|---|
| SafeBench | ✓ | ✓ | ✓ | ✓ | CARLA | ✓ |
| Scenario Runner [29] | × | ✓ | ✓ | ✓ | CARLA | × |
| DI-Drive Casezoo [30] | × | ✓ | ✓ | ✓ | CARLA | ✓ |
| SUMMIT [31] | × | × | ✓ | × | UE4 | ✓ |
| Scenario Studio [32] | × | × | × | ✓ | SMARTS | ✓ |
| CommonRoad [33] | × | × | ✓ | ✓ | None | × |
| CausalCity [34] | × | × | ✓ | ✓ | UE4 | × |
| MetaDrive [35] | × | × | ✓ | ✓ | Panda3D | ✓ |
| highway-env [36] | × | × | × | ✓ | None | × |
| SUMO NETEDIT [37] | × | × | × | ✓ | SUMO | × |
| SimMobilityST [38] | × | × | × | ✓ | None | × |
| L2R [39] | × | × | ✓ | ✓ | UE4 | × |
| AutoDRIVE [40] | × | × | ✓ | ✓ | Unity | × |
| Deepdrive [41] | × | × | ✓ | ✓ | UE4 | × |
| esmini [42] | × | × | ✓ | ✓ | Unity | × |
| AutonoViSim [43] | × | × | ✓ | ✓ | PhysX | × |

- **Diverse metrics on safety measurement of different AD algorithms**. We report our evaluation based on 10 evaluation metrics, based on three levels: safety, functionality, and etiquette.

- **General leaderboard of safety evaluation and extensible findings**. We provide a comprehensive leaderboard for the robustness and safety evaluation of 4 AD algorithms, and we observe different performances of these AD algorithms under different controllable settings.

- **High flexibility and effectiveness**. Our evaluation platform is flexible to be integrated into other simulation platforms and different devices. Once the AD algorithm is trained, it is very effective to be tested on our generated testing scenarios.

## 2 Related work

Existing AD algorithm evaluation approaches and platforms can be categorized into three types based on how the testing driving scenarios are generated. First, the **data-driven** based generation and testing approaches [44, 45, 46, 47] focus on real-world data sampling and distribution density estimation. This line of research is able to model the real-world driving conditions, while requiring a large number of data collection to capture the "rare" safety-critical scenarios for testing. Second, the **adversary-based** generation and testing approaches [48, 49, 50] model the surrounding agents (e.g., vehicles and pedestrians) as adversarial agents to generate safety-critical driving scenarios. Third, the **knowledge-based** generation and testing approaches [51, 33, 52] aim to integrate domain knowledge such as traffic rules as additional constraints to guide the testing scenario generation process. Recently, the latter two categories have shown efficient and effective evaluation results under specific driving environments and settings, and therefore we mainly focus on them in this work. However, existing driving scenario generation and testing approaches are developed on different platforms with different AD algorithms and sensor configurations, etc., making it challenging to directly compare the effectiveness of different testing scenarios, scenario generation algorithms, and the safety of AD algorithms. Thus, in this work we will provide the *first* unified platform SafeBench, to generate safety-critical scenarios with different algorithms considering a range of environments and configurations for fair comparison based on a comprehensive set of evaluation metrics. In addition, several works have been conducted to test the safety of autonomous vehicles from the software testing perspective [53], which mainly focuses on identifying the safety violations from the software level. Such testing frameworks can be integrated into SafeBench as well for comprehensive testing.

**Comparison with other AD evaluation platforms** To accurately posit our SafeBench platform in the AD evaluation area, we summarize existing platforms developed for autonomous vehicle evaluation and compare them with our platform in Table 1. We notice that very few of them consider safety-critical scenarios and the number of scenarios in existing evaluation platforms is very limited.

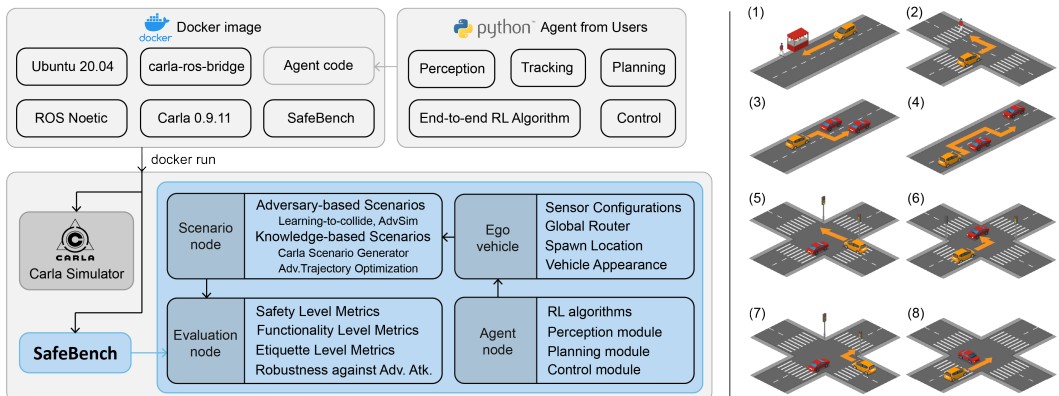

Figure 1: **Left:** Framework overview of SafeBench. **Right:** 8 safety-critical driving scenarios - (1) Straight Obstacle (2) Turning Obstacle (3) Lane Changing (4) Vehicle Passing (5) Red-light Running (6) Unprotected Left-turn (7) Right-turn (8) Crossing Negotiation.

# 3 SafeBench: benchmarking platform for safety evaluation

In this section, we will first provide an overview of our platform SafeBench, followed by the details of our developed scenario generation algorithms and variants, as well as the evaluation metrics.

## 3.1 Platform structure

**Overview.** In Figure 1, we show the structure of our SafeBench platform. This platform runs in the Docker [54] container and is built upon the Carla simulator [28]. We use ROS [55] for communication between the modules in the platform. In particular, SafeBench consists of 4 components (nodes) as introduced in the following.

**Ego vehicle** provides a virtual vehicle including the configurations of sensors (e.g., the positions and parameters of LiDAR, Camera, and Radar), the global planner, and the appearance of the vehicle. The testing AD algorithms are deployed in this node to interact with the driving scenarios. Users can change the configuration of this node to satisfy the requirement of their algorithms.

**Agent node** is designed to train and manage AD algorithms for ego and surrounding vehicles, taking as input the observation information from the testing scenarios and outputting the controlling signals. AD algorithms managed by this node can be trained on our platform.

**Scenario node** is the core part of SafeBench, which is responsible for organizing and generating testing scenarios. These scenarios control the behaviors of traffic participants (e.g., pedestrians and surrounding vehicles) and static driving environments (e.g., road layout and status of traffic lights).

**Evaluation node** is designed to provide comprehensive evaluations by testing different AD algorithms under diverse generated driving scenarios based on different metrics. The Evaluation Node collects all information during testing and provides an evaluation summary on different levels.

## 3.2 Safety-critical testing scenarios

In this section, we first define the safety-critical traffic testing scenarios we considered in this work, containing 8 most representative and challenging driving scenarios of pre-crash traffic [24] summarized by the National Highway Traffic Safety Administration (NHTSA). In addition, for each scenario, we design ten diverse driving routes that vary in terms of surrounding environments, number of lanes, road signs, etc. Please see more detailed scenario definitions and route variants in Appendix A.3.

**Pre-crash safety-critical scenarios.** We show the 8 pre-crash scenarios in the right part of Figure 1. In each scenario, the ego vehicle needs to drive along a pre-defined route and react to emergencies that occur on the road while driving. Throughout the process, the ego vehicle should follow the traffic rules and avoid potential car accidents.

**Driving routes.** In practice, a driving scenario may involve many variants. For instance, small changes in the vehicle location or in the surrounding environment may lead to big changes in vehicle decision-making. In order to provide a more comprehensive safety evaluation, we design 10 driving routes for each safety-critical scenario. Each driving route has a sequence of pre-defined waypoints. Different driving routes of the same scenario may have a different number of lanes, different scenes (e.g., intersections, T-junctions, bridges, etc.), or different road signs, which restrict vehicle behaviors in different ways. We show 2 example route variants of *Turning Obstacle* in Figure 2.

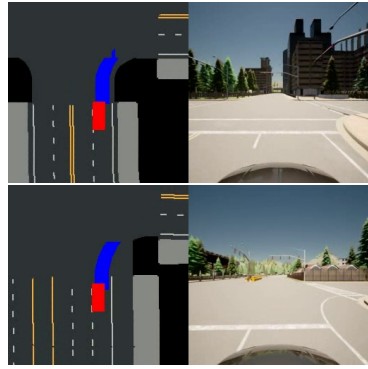

Figure 2: An example of route variants in *Turning Obstacle* scenario, consisting of a different number of lanes (2-lane vs. 3-lane road) and surrounding buildings.

### 3.3 Safety-critical scenario generation algorithms

In this section, we detail how we collect and optimize safety-critical testing scenarios using different generation algorithms. Specifically, for each driving route mentioned above, we develop 4 algorithms to generate various testing samples. These algorithms mainly fall into two categories: adversary-based generation and knowledge-based generation.

#### 3.3.1 Adversary-based generation

The state-of-the-art adversarial generation algorithms usually consist of two components: the scenario generator, and the victim model (i.e., the ego vehicle or tested AD agent). Existing adversarial generation frameworks adopt different strategies to manipulate traffic scenarios, such as perturbing the position of surrounding vehicles (SVs) or forcing a cyclist to take an adversarial action, such that the victim model will crash into SVs and fail in the generated scenario. To examine the safety and robustness of the tested AD agent against such adversarial scenarios, we select two representative algorithms as follows: ($i$) **Learning-to-collide (LC)** [56] is a black-box algorithm that optimizes the initial poses of a cyclist to attack the AD algorithm. Following the default setting, we formulate the traffic scenarios as a series of auto-regressive building blocks and obtain the generated scenarios by sampling from the joint distribution of these blocks. The policy gradient method REINFORCE [57] is used to solve the scenario optimization problem. In LC, the authors only focus on generating *Turning Obstacle* scenario, so we adapt the method to all the 8 scenarios and generate different initial conditions for all the driving routes. ($ii$) **AdvSim (AS)** [25] directly manipulates existing trajectories to perturb the driving paths of SVs, posing dangers to the tested AD agent. We follow the default setting and use the kinematic bicycle model [58] to represent and calculate the full trajectory of SVs. Based on the results obtained by interacting with the driving environment, we optimize the trajectory parameters using the black-box search algorithm Bayesian Optimization [59, 60]. Similarly, in our experiments, we generate adversarial trajectories for all the route variants.

#### 3.3.2 Knowledge-based generation

In the physical world, driving scenarios need to satisfy traffic rules and physical laws. Scenarios generated by adversarial algorithms, however, sometimes violate these rules. Therefore, we develop novel generation algorithms that integrate domain knowledge into the generation process. We select two representative algorithms as follows. ($i$) **Carla Scenario Generator (CS)** [29] is a module built on the Carla Simulator [28] which uses rule-based methods to construct testing scenarios. Following the standard process, we adopt the rules and use grid search to generate safety-critical scenario parameters for all the 8 traffic scenarios. ($ii$) **Adversarial Trajectory Optimization (AT)** [49] uses explicit knowledge as constraints to guide the scenario optimization process. We adopt the same constraints that needed to be satisfied and use the default PSO-based [61] blackbox optimization for generating all kinds of testing scenarios in SafeBench.

### 3.4 Evaluation metrics

In this section, we introduce the evaluation metrics used in SafeBench. Specifically, we evaluate the performance of AD algorithms on 3 levels: *Safety level*, *Functionality level*, and *Etiquette level*. Within each level, we design several metrics focusing on different aspects. Finally, an *overall score* is calculated as a weighted sum of all the evaluation metrics introduced below.

**Safety level**  To evaluate the safety of given AD algorithms, we follow existing works [62, 63] and consider 4 evaluation metrics focusing on serious violations of traffic rules: *collision rate* (CR), *frequency of running red lights* (RR), *frequency of running stop signs* (SS), and *average distance driven out of road* (OR). Formally, we define the scenario trajectory as $\tau$, which is sampled from a scenario distribution $\mathcal{P}$, then the collisions that happened in one scenario after testing the AD algorithm can be represented as $c(\tau)$. Similarly, we obtain the number of running red lights $r(\tau)$, running stop signs $s(\tau)$, and distance driven out of road $d(\tau)$. The 4 metrics are concretely calculated as: $CR = \mathbb{E}_{\tau \sim \mathcal{P}}[c(\tau)]$, $RR = \mathbb{E}_{\tau \sim \mathcal{P}}[r(\tau)]$, $SS = \mathbb{E}_{\tau \sim \mathcal{P}}[s(\tau)]$, and $OR = \mathbb{E}_{\tau \sim \mathcal{P}}[d(\tau)]$.

**Functionality level**  In each testing scenario, the AD agent is expected to follow and complete a specific route. This level of evaluation metrics is used to measure the functional ability of AD agents to finish such a task. Inspired by previous works [62, 63], we develop 3 metrics as follows: *route following stability* (RF), *average percentage of route completion* (Comp), and *average time spent to complete the route* (TS). To calculate RF, we use the average distance between the ego vehicle and the reference route during each testing $x(\tau)$. Then we calculate $RF = 1 - \mathbb{E}_{\tau \sim \mathcal{P}}[\min\left\{\frac{x(\tau)}{x_{max}}, 1\right\}]$, where $x_{max}$ is a constant indicating the maximum deviation distance. Comp is calculated as $Comp = \mathbb{E}_{\tau \sim \mathcal{P}}[p(\tau)]$, where $p(\tau)$ is the percentage of route completion of each testing scenario. TS is the average time spent for completing the routes successfully: $TS = \mathbb{E}_{\tau \sim \mathcal{P}}[t(\tau)|p(\tau) = 100\%]$, where $t(\tau)$ denotes the time cost of each testing scenario.

**Etiquette level**  In practice, driver etiquette is an indicator of the driving skills of AD algorithms. Here we follow existing works [64, 65] and consider 3 metrics accordingly: *average acceleration* (ACC), *average yaw velocity* (YV), and *frequency of lane invasion* (LI). Similarly, these metrics are calculated as the expectation over all testing scenarios: $ACC = \mathbb{E}_{\tau \sim \mathcal{P}}[acc(\tau)]$, $YV = \mathbb{E}_{\tau \sim \mathcal{P}}[y(\tau)]$, and $LI = \mathbb{E}_{\tau \sim \mathcal{P}}[l(\tau)]$, where $acc(\tau), y(\tau), l(\tau)$ denote the accelerations, yaw velocities, and number of lane invasions respectively.

**Overall score**  To obtain an evaluation overview of the quality of AD algorithms, we aggregate all the metrics and report an *overall score* (OS), which is a weighted sum of the 10 metrics introduced above. Specifically, the overall score is calculated as: $OS = \sum_{i=1}^{10} w^i \times g(m^i)$, where $m^i$ is the $i^{th}$ metric, $w^i$ is the corresponding weight, $g(m^i)$ is defined as

$$g(m^i) = \begin{cases} \frac{m^i}{m^i_{max}}, & m^i \text{ is the higher the better} \\ 1 - \frac{m^i}{m^i_{max}}, & m^i \text{ is the lower the better} \end{cases} \tag{1}$$

where $m^i_{max}$ is a constant indicating the maximum allowed value of $m^i$. More details of the constant, parameters, and weight selection are in Appendix A.4.

## 4  Benchmark evaluation on SafeBench

In this section, we will first introduce the AD algorithms we will test which are based on different input state types, then illustrate our testing scenario generation and selection details, followed by our comprehensive benchmark results and corresponding observations and findings.

### 4.1  AD algorithms tested on SafeBench

We test various types of algorithms based on the safety-critical scenarios in SafeBench. We particularly focus on reinforcement learning-based self-driving methods, since they require minimum domain knowledge of the overall system and driving scenarios [66, 67, 68, 69]. One only needs to specify the reward function, action space, and state space, then train the agent by interacting with the scenario, and finally obtain a self-driving agent with reasonable performance. The reward function is given by a linear combination of the route following bonus, the collision penalty, the speeding penalty, and the energy consumption penalty. The action space is specified by the steering and throttle of the vehicle.

We select 4 representative deep RL methods for evaluation, including a stochastic on-policy algorithm – Proximal Policy Optimization (PPO) [70], a stochastic off-policy method – Soft Actor-Critic (SAC) [71], and two deterministic off-policy approaches – Deep Deterministic Policy Gradient

(DDPG) [72] and Twin Delayed DDPG (TD3) [73]. To encourage the diversity of evaluation agents, we vary the state space to equip them with different perceptual capabilities. We design 4 **state spaces** for each RL algorithm based on previous works [67, 68] as follows. The detailed model design and hyperparameters are presented in appendix A.5.

- **4D**. The basic observation type contains only 4 dimensions of observation: distance to the waypoint, longitude speed, angular speed, and a front-vehicle detection signal.

- **4D+Dir**. For a more complex observation type, we add another 7 dimensions of observations, which are "Command (turn left, turn right or go straight)" and vectors that represent the direction of the ego vehicle, current waypoint, and target waypoint.

- **4D+BEV**. We render the ego vehicle's local semantic map using the information provided by CARLA as the bird's-eye view (BEV) image, where the vehicles are represented by boxes. Lanes and routes are represented by line segments. We incorporate the BEV image together with 4 dimensional states to form this observation type.

- **4D+Cam**. This observation type includes an image captured by the front camera with 4D.

## 4.2 Driving scenarios for testing

**Scenario generation.** We apply 4 safety-critical scenario generation algorithms to 8 template scenarios, each of which contains 10 diverse driving routes. For each generation algorithm, we keep 9 or 10 testing scenarios based on their qualities. Thus, in total, we generate 3,140 testing scenarios for evaluation. We note that some scenario generation algorithms require a surrogate model to search for effective safety-critical configurations. For instance, we follow the setup of LC [56] to train a surrogate SAC model based on random benign scenarios.

**Scenario selection.** After collecting the raw testing scenarios, we select scenarios with desired properties. Specifically, we test all the generated scenarios on 4 AD algorithms with basic observation type and select scenarios that cause the most collisions. In Figure 3, we show a histogram of the distribution for collisions. We only keep scenarios that cause collisions for at least 2 algorithms during testing, which is shown in red in Figure 3. The selected testing scenarios have high transferability across AD algorithms and high risk levels, which further improves both the effectiveness and efficiency of AD evaluation. After the selection, we obtain 2,352 testing scenarios in total. More details can be found in Appendix A.1.

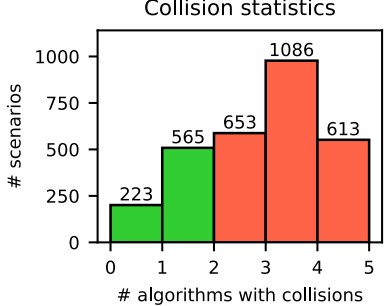

Figure 3: Collision statistics of generated scenarios before scenario selection. Red bars represent the selected ones with high collision rate. Green bars represent the unselected scenarios with low collision rate.

**Analysis of generation algorithms and testing scenarios.** We analyze the properties of scenario generation algorithms based on a range of metrics, including the *collision rate* (CR), *overall score* (OS), and the overall *selection rate* (SR) for each scenario before and after selection. As shown in Table 2, first, the scenario selection process indeed helps to improve CR of the testing scenarios to induce more safety-critical ones: with the highest improvement as 30% for LC. Second, AT is the most effective algorithm to cause both high CR and low OS. In fact, 73.4% of the generated scenarios by AT can cause collisions to the surrogate model, and it will increase to 81.1% after scenario selection. The scenarios generated by AT achieve OS as 0.546, and it will further decrease to 0.508 after scenario selection, indicating its testing effectiveness. Third, regarding the overall SR of different algorithms, scenarios generated by CS achieve the highest SR, which means CS is the best algorithm in terms of transferability across different AD algorithms. Specifically, 85.5% of scenarios generated by CS can successfully cause collisions to other unseen AD agents. Finally, among different scenarios, *Vehicle Passing* is the most difficult with the highest CR and lowest OS.

## 4.3 Benchmark results

We train our AD algorithms on random benign scenarios and evaluate them on SafeBench. We present the training details in Appendix A.6 and we provide important findings in the following.

Table 2: **Statistics of scenario generation/selection**. We report *collision rate* (CR) before and after scenario selection (S-CR) to measure the effectiveness of different scenario generation algorithms. The *overall score* (OS) before and after scenario selection (S-OS) are used to demonstrate the safety-critical scenario generation capability of different algorithms. The *selection rate* (SR) is reported to evaluate the transferability of generation algorithms across AD agents. The last column shows the average over all the scenarios, with bold numbers indicating the best performance among the 4 generation algorithms. LC: Learning-to-collide, AS: AdvSim, CS: Carla Scenario Generator, AT: Adversarial Trajectory Optimization, ↑/↓: higher/lower the better.

| Metric | Algo. | Traffic Scenarios | | | | | | | | Avg. |
| | | Straight Obstacle | Turning Obstacle | Lane Changing | Vehicle Passing | Red-light Running | Unprotected Left-turn | Right-turn | Crossing Negotiation | |
|---|---|---|---|---|---|---|---|---|---|---|
| CR ↑ | LC | 0.320 | 0.140 | 0.560 | 0.920 | 0.410 | 0.630 | 0.458 | 0.470 | 0.489 |
| | AS | 0.570 | 0.350 | 0.650 | 0.900 | 0.600 | 0.820 | 0.520 | 0.550 | 0.620 |
| | CS | 0.610 | 0.630 | 0.322 | 0.900 | 0.767 | 0.756 | 0.667 | 0.711 | 0.670 |
| | AT | 0.680 | 0.310 | 0.700 | 0.930 | 1.000 | 0.850 | 0.500 | 0.900 | **0.734** |
| S-CR ↑ | LC | 0.756 | 0.923 | 0.560 | 0.919 | 0.833 | 0.870 | 0.661 | 0.793 | 0.789 |
| | AS | 0.794 | 0.595 | 0.650 | 0.900 | 0.833 | 0.930 | 0.792 | 0.797 | 0.787 |
| | CS | 0.967 | 0.684 | 0.322 | 0.900 | 0.932 | 0.870 | 0.711 | 0.797 | 0.773 |
| | AT | 0.847 | 0.485 | 0.697 | 0.930 | 1.000 | 0.966 | 0.562 | 1.000 | **0.811** |
| OS ↓ | LC | 0.765 | 0.825 | 0.613 | 0.451 | 0.755 | 0.632 | 0.630 | 0.646 | 0.665 |
| | AS | 0.654 | 0.718 | 0.577 | 0.465 | 0.659 | 0.544 | 0.599 | 0.606 | 0.603 |
| | CS | 0.629 | 0.577 | 0.738 | 0.464 | 0.569 | 0.571 | 0.520 | 0.522 | 0.574 |
| | AT | 0.600 | 0.737 | 0.557 | 0.455 | 0.460 | 0.526 | 0.607 | 0.423 | **0.546** |
| S-OS ↓ | LC | 0.565 | 0.461 | 0.613 | 0.451 | 0.533 | 0.518 | 0.528 | 0.476 | 0.518 |
| | AS | 0.548 | 0.600 | 0.577 | 0.465 | 0.535 | 0.492 | 0.451 | 0.480 | 0.518 |
| | CS | 0.465 | 0.550 | 0.738 | 0.464 | 0.483 | 0.519 | 0.496 | 0.473 | 0.524 |
| | AT | 0.523 | 0.654 | 0.558 | 0.455 | 0.460 | 0.471 | 0.574 | 0.372 | **0.508** |
| SR ↑ | LC | 0.410 | 0.130 | 1.000 | 0.990 | 0.420 | 0.690 | 0.590 | 0.580 | 0.601 |
| | AS | 0.680 | 0.420 | 1.000 | 1.000 | 0.720 | 0.860 | 0.530 | 0.640 | 0.731 |
| | CS | 0.600 | 0.760 | 1.000 | 1.000 | 0.822 | 0.856 | 0.922 | 0.878 | **0.855** |
| | AT | 0.590 | 0.330 | 0.990 | 1.000 | 1.000 | 0.870 | 0.890 | 0.900 | 0.821 |

Table 3: **The performance of AD algorithms on SafeBench**. We report the average *overall score* (OS) on testing scenarios generated by all the 4 scenario generation algorithms with driving route variations. *Benign* indicates the performance of AD algorithms tested on normal driving scenarios. The last two columns show the OS averaged over all benign and safety-critical traffic scenarios.

| Model | Traffic Scenarios | | | | | | | | Avg. Benign | Avg. Safety-critical |
| | Straight Obstacle | Turning Obstacle | Lane Changing | Vehicle Passing | Red-light Running | Unprotected Left-turn | Right-turn | Crossing Negotiation | | |
|---|---|---|---|---|---|---|---|---|---|---|
| DDPG (4D) | 0.545 | 0.526 | 0.440 | 0.501 | 0.611 | 0.444 | 0.411 | 0.507 | 0.603 | 0.498 |
| SAC (4D) | 0.533 | 0.474 | **0.577** | 0.471 | 0.482 | 0.501 | 0.503 | 0.432 | **0.833** | 0.497 |
| TD3 (4D) | 0.479 | 0.596 | 0.477 | **0.592** | 0.532 | 0.525 | 0.459 | 0.482 | 0.830 | 0.518 |
| PPO (4D) | **0.761** | **0.611** | 0.426 | 0.432 | **0.755** | **0.728** | **0.605** | **0.655** | 0.819 | **0.622** |

**Performance of AD on benign and safety-critical scenarios.** The benchmark results of AD algorithms based on 4D inputs are summarized in Table 3. From the table, we observe a large performance gap in AD algorithms tested on benign and safety-critical scenarios in SafeBench. For example, although TD3 achieves an overall score of 0.830 on benign scenarios, it only achieves 0.518 when testing on safety-critical scenarios. In general, agents that perform well in benign scenarios usually fail given the safety-critical ones, indicating a trade-off between the performance under benign and safety-critical testing scenarios. For instance, PPO obtains the highest overall score on safety-critical scenarios, while its benign performance is worse than both SAC and TD3. On the other hand, although SAC achieves the highest overall score on benign testing scenarios, its performance under safety-critical ones is the worst. More results on algorithms with other types of input observations can be found in Appendix A.8.

**Comprehensive diagnostic report of AD algorithms in all scenarios.** In order to provide a comprehensive understanding of the performance of AD algorithms, we conducted a detailed diagnostic

Table 4: **Diagnostic report**. We test every AD algorithm on all selected testing scenarios and report the evaluation results on three different levels. CR: collision rate, RR: frequency of running red lights, SS: frequency of running stop signs, OR: average distance driven out of road, RF: route following stability, Comp: average percentage of route completion, TS: average time spent to complete the route, ACC: average acceleration, YV: average yaw velocity, LI: frequency of lane invasion, OS: overall score, ↑/↓: higher/lower the better.

| Model | Safety Level | | | | Functionality Level | | | Etiquette Level | | | OS ↑ |
|---|---|---|---|---|---|---|---|---|---|---|---|
| | CR ↓ | RR ↓ | SS ↓ | OR ↓ | RF ↑ | Comp ↑ | TS ↓ | ACC ↓ | YV ↓ | LI ↓ | |
| DDPG (4D) | 0.780 | **0.089** | **0.087** | 12.619 | 0.504 | 0.466 | 20.860 | 2.488 | **0.405** | 5.764 | 0.489 |
| SAC (4D) | 0.829 | 0.216 | 0.146 | 3.115 | 0.882 | 0.648 | **16.827** | **1.830** | 0.704 | 2.580 | 0.499 |
| TD3 (4D) | 0.783 | 0.231 | 0.141 | 2.535 | **0.903** | 0.670 | 17.644 | 2.680 | 1.493 | **2.545** | 0.516 |
| PPO (4D) | **0.603** | 0.287 | 0.150 | **0.099** | 0.901 | **0.751** | 18.021 | 2.461 | 1.506 | 3.528 | **0.606** |

Table 5: **Robustness of point cloud segmentation**. We test 4 point cloud segmentation models against three adversarial attacks and report the IoU.

| Method | PointNet++ [77] | SqueezeSeg [78] | PolarSeg [79] | Cylinder3D [80] |
|---|---|---|---|---|
| Benign | $0.81 \pm 0.01$ | $0.82 \pm 0.01$ | $0.95 \pm 0.01$ | $0.96 \pm 0.01$ |
| Point Attack | $0.80 \pm 0.01$ | $0.82 \pm 0.02$ | $0.94 \pm 0.01$ | $0.96 \pm 0.01$ |
| Pose Attack | $0.40 \pm 0.08$ | $0.47 \pm 0.04$ | $0.89 \pm 0.01$ | $0.88 \pm 0.02$ |
| Scene Attack | $0.52 \pm 0.12$ | $0.65 \pm 0.04$ | $0.85 \pm 0.01$ | $0.86 \pm 0.01$ |

report for each tested algorithm from different perspectives. In particular, we consider three levels of evaluation metrics: Safety, Functionality, and Etiquette, as shown in Table 4 for the 4D-based AD agents. Comprehensive reports of all AD agents are in Appendix A.9. We observe that different AD algorithms outperform others under different metrics. For instance, on the *Safety level*, PPO achieves the lowest CR and OR, which means it has a high level of safety and a low accident rate, while its performance on the Etiquette level is relatively low. On the *Functionality level*, TD3 achieves the highest route following stability, demonstrating its ability to complete given tasks without deviating from the route. On the *Etiquette level*, SAC and DDPG achieve the lowest ACC and YV respectively, which measure the driving quality. Based on the overall score (OS), PPO is shown to be the best AD algorithm given the weighted average over all metrics.

We also notice a trade-off between functionality-level metrics and safety-level metrics. From Table 4, we can observe that an agent with strong functionality performance may not be safe regarding the safety level metrics. For instance, the SAC agent achieves the best TS score, which means that it can finish the routes in the shortest time, but its collision rate (CR) is also the highest among all the other agents. Similarly, the PPO agent that achieves the best route completion (Comp) score presents, however, the highest RR and SS scores, which means that it may run red lights and stop signs most frequently. This observation suggests the inherent contradiction between some safety metrics and functionality metrics, which is also unveiled in some previous studies [74, 75, 76].

## 4.4 Robustness Evaluation: Physical semantic attacks against AD algorithms

With our modularized design, SafeBench is also able to identify vulnerabilities of different components in AD systems by performing diverse *adversarial attacks* in addition to the tests on safety-critical scenarios. Here we provide evaluations against physical semantic attacks on perception components in AD such as LiDAR and multi-sensor systems, considering both the semantic segmentation and 3D object detection tasks. More evaluations and visualizations are in Appendix A.10.

**Point cloud segmentation**    To test the robustness of point cloud segmentation in AD, we implement 3 types of adversarial attacks: (1) *Point Attack*: a point-wise attack method [4] that adds small disturbance to the positions of 3D points; (2) *Pose Attack*: a scene generation method that searches for adversarial poses of vehicles; (3) *Scene Attack*: a semantically controllable generative method based on SAG [51]. For Pose attack and Scene Attack, we first generate the locations and orientations of vehicles and spawn them in Carla. Then, we use the LiDAR sensor to collect point clouds needed by the segmentation algorithms. We select 4 segmentation models (PointNet++ [77], PolarSeg [79], SqueezeSegV3 [78], Cylinder3D [80]) as our victim models, all of which are pre-trained on Semantic

Table 6: **Robustness of 3D object detection.** We report the average precision (AP) of car class for models taking 3D LiDAR point clouds and multi-modal data as inputs respectively.

| Data Source | Model | Input Data | 3D AP (%) | | | Bird's Eye View AP (%) | | |
|---|---|---|---|---|---|---|---|---|
| | | | easy | moderate | hard | easy | moderate | hard |
| Benign | SECOND | LiDAR | **87.31** | **86.81** | **86.81** | **88.95** | **88.92** | **88.92** |
| | CLOCs | LiDAR+Img | 76.90 | 76.50 | 76.50 | 81.73 | 82.01 | 82.01 |
| Adversarial | SECOND | LiDAR | 60.74 | 59.81 | 59.81 | 67.87 | 63.64 | 63.64 |
| | CLOCs | LiDAR+Img | **61.98** | **61.98** | **61.98** | **76.64** | **76.64** | **76.64** |

Kitti dataset [81]. We present the evaluation results of IoU after attacks in Table 5. The results show that all 4 models can be attacked and have very different performances under attacks. This demonstrates the ability of SafeBench on evaluating point cloud perception task in AD systems.

**3D object detection**  To evaluate the robustness of recognizing and locating surrounding objects for AD systems, we perform different attacks on 3D object detection task. We place the AD agent in a fixed scene and put different objects such as vehicles, pedestrians, and other traffic objects in front of the agent to test the detection accuracy. We follow TSS [82] to perform adversarial physical semantic transformations to both camera image and LiDAR point clouds. We incorporate 4 kinds of semantic transformations to attack the perception component in AD systems. Specifically, we first consider changing different types of vehicles such as *Tesla Model 3*, *Audi TT*, and *Nissan Patrol*. Second, we perturb the color of each vehicle. We choose 5 most common colors to test the robustness of AD algorithms and any RGB value can be applied to the car in SafeBench. Third, we change different properties for pedestrians, such as body shapes and skin colors. Finally, we perform rotation on every object to examine the reliability of AD systems. We present our results in Table 6. We train different 3D object detection models on normal driving scenarios and test the models on adversarial data generated by SafeBench. The SECOND [83] takes LiDAR point clouds as input while CLOCs [84] is a multi-modal model that takes both LiDAR point clouds and camera images as inputs. From Table 6, we find that the SECOND model performs better on benign data. However, on adversarial data, CLOCs achieves higher average precision and the performance drop of the CLOCs model from benign to adversarial is much smaller than that of the SECOND model. One reason could be that data from both modalities complement each other, helping the model to make better decisions, which indicates potential designing strategies for AD algorithms: use multi-sensor fusion models to incorporate and process multi-modal data, leading to higher robustness. This demonstrates the ability of SafeBench to evaluate the robustness of object detection task.

## 5   Conclusion

In this paper, we introduce SafeBench, the first unified platform to automatically evaluate and analyze the performance of AD algorithms in multiple aspects using various safety-critical driving scenarios generated by different generation algorithms. We incorporate 8 safety-critical scenarios and 10 evaluation metrics from 3 different levels to provide a detailed diagnostic report for each AD agent. AD algorithms tested on SafeBench have a large performance drop compared to evaluations on benign scenarios, suggesting the deficiencies of each algorithm and the effectiveness of our testing platform. We hope our platform and findings will serve as a reliable and comprehensive benchmark to help researchers and practitioners to identify weaknesses in existing AD systems and further develop safe AD algorithms as well as more effective testing scenario generation algorithms.

**Limitations**  Although simulation is a useful and necessary tool for evaluating AD systems given its efficiency and controllability [85, 86], the simulation in SafeBench cannot exactly reflect real-world conditions. On-track testing is necessary before deploying AD algorithms in the real world. Besides, we only evaluate RL-based AD algorithms in the current version of SafeBench, and testing more diverse AD algorithms, including commercial systems such as Baidu Apollo [87] and Openpilot [88] would be interesting future work.

## Acknowledgments and Disclosure of Funding

This work is partially supported by the NSF grant No.1910100, NSF CNS No.2046726, C3 AI, and the Alfred P. Sloan Foundation.

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
