# OpenReview forum: "SafeBench: A Benchmarking Platform for Safety Evaluation of Autonomous Vehicles"
_NeurIPS.cc/2022/Track/Datasets_and_Benchmarks — NeurIPS 2022 Datasets and Benchmarks _

### Official Review · Reviewer_pkfV · 2022-07-12
**Minor comments for a well-written paper**

**Rating:** 7
**Confidence:** 4
**Correctness:** Yes

**Strengths:**

The paper is well written and the contributions are clearly presented.
The work is novel and could represent a step forward in the assessment of the safety of autonomous driving.
The benchmark is overall well designed and fairly comprehensive, and the main project page includes additional videos and the learderboard.


**Weaknesses:**

The paper presents a few weaknesses that could be easily solved:
- there is no description of how the traffic scenarios are generated during training of the agent. Additional details could be reported in the supplementary material;
- there is no description of how the BEV images are obtained. In the real world, these are typically obtained with neural networks or other perception pipelines processing images/pointclouds. If this pipeline is considered as "perfect", and the BEV images are collected directly in CARLA, then it is better to highlight this point;
- I feel like the Overall Score could be misleading, depending on the weights that are applied. Maybe it could be an idea to also consider 3 different overall scores to sum up the safety, functionality, and etiquette levels.
- The choice of the 8 scenarios selected from the NHTSA documents should be justified, since the scenarios listed are way more. Were these scenarios selected by statistical frequency on the roads or by other metrics?
- It is not clear how to submit additional results to the leaderboard.

**Additional Feedback:**

In my opinion, this is a paper that is worth acceptance for technical contributions, clarity, and presentation.
There are a few minor issues that can be discussed to improve the benchmark, such as summary metrics for each level, or additional clarifications.
Also, additional safety-critical scenarios should be added progressively to obtain an exhaustive assessment.

**Clarity:**

The paper is well written. There are only a few sentences that require improvements:
- lines 53-54: "previous work can be realized"
- line 246: "trains"
- line 271: "9 \sim 10" (is it 9 or 10?)

**Documentation:**

Yes, code and instructions are available.

**Relation To Prior Work:**

Yes

**Summary And Contributions:**

The paper presents a unified framework for the assessment of the safety levels of autonomous driving algorithms against 8 different safety-critical scenarios. The scenarios are generated on the CARLA simulator with 4 different generation methods and are evaluated with 10 metrics. The code and the leaderboard is provided at the dedicated website.

---

> ### Author Response · Authors · 2022-08-17
> **Thanks for your valuable comments**
>
> We thank the reviewer for recognizing our work novel and comprehensive and really appreciate the reviewer's suggestions to help improve the quality of our work. We answered the questions below and improved our paper following the suggestions.
>
> > **Q1.** There is no description of how the traffic scenarios are generated during the training of the agent. Additional details could be reported in the supplementary material.
>
> The traffic scenarios are randomly generated in benign environments during the training of the agent for generalizability. All the surrounding vehicles are operated by autopilot. We will follow your suggestions and add more detailed descriptions in the revision.
>
> > **Q2.** There is no description of how the BEV images are obtained. In the real world, these are typically obtained with neural networks or other perception pipelines processing images/point clouds. If this pipeline is considered "perfect", and the BEV images are collected directly in CARLA, then it is better to highlight this point;
>
> The BEV images used in our platform are perfect and directly collected in CARLA. We will follow your suggestions and highlight this point in the revision.
>
> > **Q3.** I feel like the Overall Score could be misleading, depending on the weights that are applied. Maybe it could be an idea to also consider 3 different overall scores to sum up the safety, functionality, and etiquette levels.
>
> We agree that using one score to indicate the AD performance is not practically possible. However, this score gives some intuition about the performance with weights designed by users. We report the weights used by the benchmark in Appendix Table 5. Users can also adjust the weights during testing according to their specific application and testing purpose. We also provide the 3 different overall scores following the suggestions in revision Appendix A.9, and we will make it more clear in the revision.
>
> > **Q4.** The choice of the 8 scenarios selected from the NHTSA documents should be justified, since the scenarios listed are way more. Were these scenarios selected by statistical frequency on the roads or by other metrics?
>
> For the choices of 8 traffic scenarios, we follow the NHTSA scenario ranking and the Carla Challenge to choose safety-critical traffic scenarios that have the highest rates of occurrence in the real world, which constitute the majority of the real-world scenarios. In addition, SafeBench is highly flexible. Users can also add customized scenarios with little effort. We will follow your suggestion and add more discussion in the revision.
>
> > **Q5.** It is not clear how to submit additional results to the leaderboard.
>
> With our modular design, users only need to provide a customized Agent Node with several basic APIs such as perception and planning in order to join the benchmark. We will provide more details on how to submit results in the documentation.
>
> > **Q6.** Additional safety-critical scenarios should be added progressively to obtain an exhaustive assessment.
>
> Thanks for the valuable suggestion. We have added several new kinds of safety-critical scenarios which are based on physical semantic adversarial attacks against sensor input in revision Section 4.4. Specifically, we include point attack [1], scene attack [2], and pose attack to generate adversarial point cloud data. We also apply semantic transformations [3] to perturb multi-modal data. We will keep adding safety-critical scenarios and improving the comprehensiveness and reliability of SafeBench.
>
> —
>
> [1] Xiang, Chong, Charles R. Qi, and Bo Li. "Generating 3d adversarial point clouds." Proceedings of the IEEE/CVF Conference on Computer Vision and Pattern Recognition. 2019.
>
> [2] Ding, Wenhao, et al. "Semantically controllable scene generation with guidance of explicit knowledge." arXiv preprint arXiv:2106.04066 (2021).
>
> [3] Li, Linyi, et al. "Tss: Transformation-specific smoothing for robustness certification." Proceedings of the 2021 ACM SIGSAC Conference on Computer and Communications Security. 2021.

---

> ### Author Response · Authors · 2022-08-24
> **Post rebuttal discussion**
>
> We sincerely thank the reviewer for your previous insightful questions and suggestions, and we have tried our best to add additional evaluations/features to our platform as well as answer the questions. Please let us know if you have further questions or comments. We really look forward to your feedback to further improve our work. Thank you!

---

> > ### Comment · Reviewer_pkfV · 2022-08-24
> > **post rebuttal discussion**
> >
> > Thank you for your revision and your comments. I do not have further comments.
> > I am keeping a rating of 7.

---

### Official Review · Reviewer_qARA · 2022-07-20
**Safebench: A Benchmarking platform for safety evaluation of autonomous vehicles**

**Rating:** 7
**Confidence:** 3
**Correctness:** No substantial factual errors were id…

**Strengths:**

- Unified test scenarios and scenario generation algorithms.
- The modularity of the platform including ego vehicle, agent, scenario and evaluation nodes, where each module could be customized based on the requirements.


**Weaknesses:**

- Lack of computation complexity analysis for scenario generation algorithms and RL based methods.
- Not enough documentation to be able to reproduce the provided results.
- Lack of generalization analysis on real-world data.


**Additional Feedback:**

- The performances for RL algorithms are only reported for the selected scenarios with high collision rates. Are we assume that they work very well for the scenarios with low collision rate?
- Any preprocessing step is done before learning the RL based method? This should be discussed in greater details.
- It seems that all the tests are performed using synthetic data. How to ensure that the learned algorithms on synthetic data could generalize as well on real data?
- In Table 1, various metrics are used to evaluate the performance of scenario generation algorithms. However, less details are provided on how these metrics are computed? Are the metrics could be used to evaluate both scenario generators and RL algorithms? If it is not the case, what overall score (OS) include for each of these steps?
- In the caption of Table 1, it is noticed that the SR (scenario rate) is reported to evaluate the transferability of generation algorithms across AD agents. More explanation should be provided.
- In section 4.3, the results are interpreted by indicating WHICH RL method works better than the others. However, it will be helpful to explain WHY one algorithm shows a better performance with regard to a specific scenario (is the reward function is defined in a way that it encourages more safety than functionality for example?)
- The authors should provide details concerning the algorithmic complexity and computation time of scenario generation algorithms as well as the RL based methods. There could be a trade-off between the complexity and precision. As it has been mentioned among the limitations of this work.
- The Nvidia-docker is used on top of the Docker environment. It implies having access to Nvidia GPU cards. Some clarification should be provided at this level. Which graphic cards are compatible and which algorithms use Cuda capability (only Deep RL methods?). The specification of the docker container, library versions, etc (this information could be added in Github readme or other online documentations).
- The authors should add a section “Datasheet for Dataset” as it has been mentioned in submission instructions for new datasets. It could clarify the nature of the used data and facilitate its comprehension.
- No indication on how to propose new RL algorithms (with a different architecture) and specific reward functions.

> I have raised the score to 7 after more in depth investigation concerning my comments

**Clarity:**

The paper is mostly clear and well written, though a few items could be improved (identified below).

**Documentation:**

- The benchmark is available at a Github repo and a Github page including experiments for different scenarios.
- It is provided how to install the platform. However, the documentation concerning how to use the platform is missing. It will be valuable to have some notes to how to run a scenario (providing readthedocs.org page for example).
- It is not easy to find out how to access the data used for the experiments.
- The appendix section misses a datasheet for dataset section


**Ethics:**

No ethical concerns identified

**Relation To Prior Work:**

The related work section provides a categorization of state-of-the-art work on the basis of how the testing driving scenarios are generated. However, it lacks a more comprehensive comparison with other similar works to precise which algorithms or functionalities are integrated in the proposed uniform platform (a summary table could be helpful).

**Summary And Contributions:**

This paper suggests a unified framework to integrate different types of safety critical testing scenarios and scenario generation algorithms with the aim to ease the comparison and the understanding of their effectiveness. Four different RL algorithms are also compared.

---

> ### Author Response · Authors · 2022-08-17
> **Thanks for your valuable comments (1/2)**
>
> We thank the reviewer for recognizing our work useful and easy to use and really appreciate the reviewer's suggestions to help improve the quality of our work. We answered the questions as below and improved our paper following the suggestions.
>
> > **Q1.** Lack of generalization analysis on real-world data. How to ensure that the learned algorithms on synthetic data could generalize as well on real data?
>
> Thanks for the insightful comments. We agree that simulation in SafeBench does not exactly reflect real-world conditions. Closing the gap between the simulation and the real world has long been a challenge for existing simulators.  Nevertheless, simulation is still popular in both research and industry, and it is generally acknowledged that simulation can be a useful tool for evaluating AD systems, given its nice efficiency and controllability [1, 2].; and simulation is an important step before the expensive real-world testing and deployment, helping researchers and developers to identify critical flaws and vulnerabilities in their AD systems before the testing in the real world, which may cause more severe consequences. We follow the standard practice to build our unified evaluation platform based on CARLA, one of the most recognized simulators. Besides, with the flexible design of SafeBench, we can train AD algorithms based on real-world data, to better strengthen the connection to the real world. We will follow your suggestions and add more discussions following the suggestions in the revision.
>
> > **Q2.** However, it lacks a more comprehensive comparison with other similar works to precise which algorithms or functionalities are integrated in the proposed uniform platform (a summary table could be helpful).
>
> Thanks for the comment. As we present the first unified platform for the safety evaluation of AD algorithms using safety-critical scenarios, there is a limited number of similar works to compare with. In particular, SafeBench is the first platform integrating comprehensive generation algorithms, safety-critical testing scenarios, and evaluation metrics, while most of the works on the evaluation platforms are not as comprehensive as SafeBench. Following the suggestion, we have added a qualitative table in revision Section 2 to summarize the comparisons between our platform and existing evaluation platforms. We will also add more related works in the revision and thanks for the suggestions!
>
> > **Q3.** It is not easy to find out how to access the data used for the experiments.
>
> Sorry for the confusion. The data is can be accessed on our GitHub repository. We will make it clear in the GitHub repo.
>
> > **Q4.** The performances for RL algorithms are only reported for the selected scenarios with high collision rates. Are we assume that they work very well for the scenarios with a low collision rate?
>
> We assume that the AD performance under scenarios with low collision rates will be better and more stable than with high collision rates (i.e., more safety-critical). This assumption is broadly accepted by the testing and adversarial machine learning community. In addition, users can still evaluate their AD algorithms on the unselected scenarios based on their needs. We will make this discussion clear in our revision.
>
> > **Q5.** Any preprocessing step done before learning the RL-based method?
>
> We follow the standard RL training process and no preprocessing step is needed before the training. The input data to RL models are directed obtained from CARLA sensors.
>
> > **Q6.** The computation of the evaluation metrics in Table 1 is confusing. Are the metrics could be used to evaluate both scenario generators and RL algorithms? If it is not the case, what overall score (OS) includes for each of these steps?
>
> The metrics can be used to evaluate both scenario generators and RL algorithms. However, they have different meanings. When evaluating scenario generators, a higher overall score indicates the generated scenarios are weaker while a lower one indicates the generated scenarios are more safety-critical. When evaluating RL algorithms, a higher overall score indicated the safety level of the AD is high, while a lower score means the algorithm is less robust and unsafe. We will make this clear in our revision.

---

> > ### Author Response · Authors · 2022-08-17
> > **Thanks for your valuable comments (2/2)**
> >
> > > **Q7.** In the caption of Table 1, it is noticed that the SR (selection rate) is reported to evaluate the transferability of generation algorithms across AD agents. More explanation should be provided.
> >
> > The selection rate measures the number of selected scenarios as a percentage of the total number of scenarios. A high selection rate means many of the scenarios generated by this generation algorithm can cause collisions to at least 2 AD algorithms, which indicates the generated scenarios have high transferability in terms of attacking different AD algorithms. We will follow your suggestion and provide more explanation in the revision.
> >
> > > **Q8.** It will be helpful to explain WHY one algorithm shows a better performance with regard to a specific scenario.
> >
> > Thanks for the insightful suggestion. SAC shows the best benign performance while PPO shows the best safety-critical overall score. One of the possible reasons is that the training process of SAC and PPO is known to be more stable than the other two algorithms [3, 4]. Besides, DDPG and TD3 are sensitive to hyper-parameters. Simply changing the hyper-parameters will lead to a huge performance drop. We will add more discussion in the revision.
> >
> > > **Q9.** Some clarification on required hardware and software should be provided.
> >
> > We ran all experiments on a desktop with one AMD 24-Core CPU, one GeForce RTX 2080 Ti GPU, and 128GB RAM. However, our platform only requires about 4GB GPU memory and 8 CPU cores, which can be satisfied in most modern desktops.
> > We will follow the suggestions to add the hardware and software requirements to our GitHub repository.
> >
> > > **Q10.** No indication on how to propose new RL algorithms (with a different architecture) and specific reward functions.
> >
> > Thanks for the insightful comment. Indeed, our evaluation platform provides recommendations on performing an overall evaluation on both normal traffic scenarios and safety-critical scenarios when proposing new algorithms. Different surroundings and environments should be considered while designing the reward function. For instance, driving in cities should get a high reward if the speed is modest, whereas driving on highways should get a high reward if the speed is high. After designing the RL algorithm, it’s simple to incorporate it in SafeBench due to our flexible modularized design. We also provide detailed instructions in our documentation.
> >
> > > **Q11.** The authors should add a section “Datasheet for Dataset” as it has been mentioned in submission instructions for new datasets.
> >
> > Thanks for the valuable comment. Since we are not proposing a new dataset, we don’t have the datasheet for the dataset. However, to help users quickly know the details of the scenarios used in our platform, we do provide a table that contains the description of all scenarios in Appendix A.3.
> >
> > > **Q12.** Lack of computation complexity analysis for scenario generation algorithms and RL-based methods.
> >
> > Thanks for the insightful comment. We follow your suggestion and add analysis for computation complexity in revision Appendix A.11.
> >
> > > **Q13.** Not enough documentation to be able to reproduce the provided results. The documentation concerning how to use the platform is missing.
> >
> > Thanks for the valuable comment. We have provided instructions to launch the SafeBench testing scenarios and reproduce our results in our GitHub repo. Since our platform is built based on CARLA, users can also follow official instructions provided by the CARLA scenario runner to customize and run a specific scenario. We will also constantly add more documentation to maintain the platform whenever we add new features.
> >
> > —
> >
> > [1] ​​Amini, Alexander, et al. "Vista 2.0: An open, data-driven simulator for multimodal sensing and policy learning for autonomous vehicles." 2022 International Conference on Robotics and Automation (ICRA). IEEE, 2022.
> >
> > [2] Thorn, Eric, et al. A framework for automated driving system testable cases and scenarios. No. DOT HS 812 623. United States. Department of Transportation. National Highway Traffic Safety Administration, 2018.
> >
> > [3] Haarnoja, Tuomas, et al. "Soft actor-critic: Off-policy maximum entropy deep reinforcement learning with a stochastic actor." International conference on machine learning. PMLR, 2018.
> >
> > [4] Haarnoja, Tuomas, et al. "Soft actor-critic algorithms and applications." arXiv preprint arXiv:1812.05905 (2018).

---

> > > ### Comment · Reviewer_qARA · 2022-08-25
> > > **Thanks for response and clarifications**
> > >
> > > Thank you for the responses and the effort made to investigate these points. The revision has mostly addressed raised concerns through my comments. I raise my score to 7.

---

> > > > ### Author Response · Authors · 2022-08-25
> > > > **Thank you for your response**
> > > >
> > > > Thank you for your response! We really look forward to further improving our work. Please let us know if you have any other feedback or suggestions. Thank you!

---

> ### Author Response · Authors · 2022-08-24
> **Post rebuttal discussion**
>
> We sincerely thank the reviewer for your previous insightful questions and suggestions, and we have tried our best to add additional evaluations/features to our platform as well as answer the questions. Please let us know if you have further questions or comments. We really look forward to your feedback to further improve our work. Thank you!

---

### Official Review · Reviewer_S2Tz · 2022-07-25
**Important topic and interesting benchmark**

**Rating:** 6
**Confidence:** 5

**Strengths:**

++ This paper presents a novel benchmark for autonomous driving.

++ The benchmark results and findings are interesting.

++ The paper is well-written and easy fo follow.

**Weaknesses:**

-- Some related work is missing.

-- Some experimental settings are not well-justified.

(See details below)

**Additional Feedback:**

Overall, I believe this is a good paper considering its novelty and contribution. I do have some questions to ask.

1. Can you add more details about your experimental settings, such as weight selection?

2. Despite the current four RL algorithms, have you ever considered some commercial practices of autonomous driving, such as Baidu Apollo and OpenPilot?

**Clarity:**

The paper is well-written.

But the first paragraph of the introduction seems confusing to me. Authors introduce a lot of adversarial sample generation for images/point clouds. While these are not strongly correlated to the paper itself, I would suggest authors shorten this part.

**Correctness:**

Overall I think the paper makes a good contribution to both the research community and industry. The benchmark seems useful, and the results are somehow interesting. I think the evaluation methods and experiments are also designed appropriately, but I also think some of the experimental settings need to be justified more carefully.

For instance, the choice of [Etiquette level] metrics. I think the evaluation from such a perspective is interesting, while I also believe it would be better if authors could justify the choice of three different metrics (any references?)

The weight selection for the overall score. While I didn't find the detailed selection criteria for these weights either in the paper or the supplementary materials. It would be nice if authors could add some details about it.

**Documentation:**

The benchmark is well-documented to support reproducibility.

**Ethics:**

No.

**Relation To Prior Work:**

I think this would be another concern. There have been a lot of attempts ([1]) regarding testing autonomous vehicles (finding safety violations) in the software engineering community. I believe they are also related to the proposed benchmark and should be discussed.

[1] Li, Guanpeng, et al. "AV-FUZZER: Finding safety violations in autonomous driving systems." 2020 IEEE 31st International Symposium on Software Reliability Engineering (ISSRE). IEEE, 2020.

[2] Gambi, Alessio, Tri Huynh, and Gordon Fraser. "Generating effective test cases for self-driving cars from police reports." Proceedings of the 2019 27th ACM Joint Meeting on European Software Engineering Conference and Symposium on the Foundations of Software Engineering. 2019.

[3] Abdessalem, Raja Ben, et al. "Testing autonomous cars for feature interaction failures using many-objective search." 2018 33rd IEEE/ACM International Conference on Automated Software Engineering (ASE). IEEE, 2018.

**Summary And Contributions:**

This paper presents SafeBench, a unified benchmarking platform for evaluating safety violations of autonomous vehicles. The proposed framework considers 8 safety-critical testing scenarios following National Highway Traffic Safety Administration and develops four different scenario generation algorithms. The benchmark includes four different deep reinforcement learning algorithms with four types of input. The benchmark results suggest that the generated scenarios of SafeBench are more challenging for current autonomous driving practices.

---

> ### Author Response · Authors · 2022-08-17
> **Thanks for your valuable comments**
>
> We thank the reviewer for recognizing our work novel and inspiring and really appreciate the reviewer's suggestions to help improve the quality of our work. We answered the questions below and improved our paper following the suggestions.
>
> > **Q1.** I think the evaluation from the etiquette level is interesting, while I also believe it would be better if authors could justify the choice of three different metrics (any references?)
>
> Thanks for the valuable suggestion! We follow several previous works to design our evaluation metrics. The *etiquette level* metrics are based on driving etiquette analysis [1, 2] which encourages AD algorithms to behave like human-driven vehicles. The *safety level* metrics and *functionality level* metrics are based on CARLA Challenge [3] and MetaDrive [4] which also evaluate the safety and reliability of AD algorithms. We summarize and modify existing widely adopted metrics to make sure every metric is useful and helpful. This also makes SafeBench more comprehensive than other single evaluation tools. We will follow your suggestions and add more discussion in the revision Section 3.4.
>
> > **Q2.** But the first paragraph of the introduction seems confusing to me. Authors introduce a lot of adversarial sample generation for images/point clouds. While these are not strongly correlated to the paper itself, I would suggest authors shorten this part.
>
> Thanks for the valuable suggestions and we have updated our revision to shorten this part to make it more clear. Thanks for the helpful comment!
>
> > **Q3.** There have been a lot of attempts regarding testing autonomous vehicles (finding safety violations) in the software engineering community. I believe they are also related to the proposed benchmark and should be discussed.
>
> Thank you very much for pointing out the references and we have added these references and corresponding discussions in the related work section in our revision.
>
> > **Q4.** The weight selection for the overall score is missing. It would be nice if the authors could add some details about it.
>
> We report the weights used by the benchmark in Appendix Table 5. Users can also adjust the weights during testing according to their specific application and testing purpose. For instance, when designing AD algorithms for children or people with disabilities, the weight of metrics in etiquette level would be higher since they may value the comfort of the vehicle more than others.
>
> In addition, to perform the safety test as in our work, we adopt two criteria for weight selection in SafeBench. First, according to the importance of each level, metrics in the safety level have the highest weights while metrics in the etiquette level have the lowest weights. Second, within the safety level, the collision rate has the highest weight since collisions can cause the most severe accident in the real world. We will follow your suggestions and add more details in the revision.
>
>
> > **Q5.** Despite the current four RL algorithms, have you ever considered some commercial practices of autonomous driving, such as Baidu Apollo and OpenPilot?
>
> Thanks for the valuable question. Our main goal is to build a comprehensive safety-critical evaluation platform for AVs. It is possible to test commercial AD systems on our platform, but we have not considered them for this initial release, as they are usually complicated, require multiple types of inputs, and are hard to draw conclusions based on the testing results. Currently, we only consider RL-based algorithms, since they require minimum domain knowledge of the overall system and driving scenarios. We will keep maintaining the platform and add more commercial AD systems for evaluation.
>
> —
>
> [1] Peng, Huei. Driving Etiquette. University of Michigan, Ann Arbor, Transportation Research Institute, 2020.
>
> [2] Huang, Xianan, Songan Zhang, and Huei Peng. "Developing robot driver etiquette based on naturalistic human driving behavior." IEEE transactions on intelligent transportation systems 21.4 (2019): 1393-1403.
>
> [3] CARLA Challenge, https://carlachallenge.org/.
>
> [4] Li, Quanyi, et al. "Metadrive: Composing diverse driving scenarios for generalizable reinforcement learning." IEEE Transactions on Pattern Analysis & Machine Intelligence 01 (2022).

---

> > ### Comment · Reviewer_S2Tz · 2022-08-24
> > **Thanks for your response**
> >
> > The revision has mostly addressed my concerns (Q1-Q4). However, I'm still concerned if the baselines are realistic enough as they are relatively simple and not close to the actual autonomous driving systems. Nevertheless, it's still an interesting work and important benchmark. I will keep the rating.

---

> > > ### Author Response · Authors · 2022-08-25
> > > **Thank you for your valuable comments**
> > >
> > > We thank the reviewer for the insightful comment. Currently, we mainly focus on RL-based AD algorithms. Testing more diverse AD algorithms, including commercial AD systems would be interesting future work. We have added a *Limitation* section in revision section 5 to reflect this following the suggestion. Thanks for the valuable feedback!

---

> ### Author Response · Authors · 2022-08-24
> **Post rebuttal discussion**
>
> We sincerely thank the reviewer for your previous insightful questions and suggestions, and we have tried our best to add additional evaluations/features to our platform as well as answer the questions. Please let us know if you have further questions or comments. We really look forward to your feedback to further improve our work. Thank you!

---

### Official Review · Reviewer_Ss1H · 2022-07-26
**A good benchmark for safety evaluation of autodriving systems**

**Rating:** 7
**Confidence:** 4
**Correctness:** Appropriate.
**Clarity:** Yes.

**Strengths:**

1. This paper proposes SafeBench, the first platform as far as I know for the safety evaluation of auto-driving systems.
2. The benchmark is modularized design and easy to use, which would be highly beneficial for researchers in this area.
3. This platform contains several safety-critical scenarios for auto driving and multiple evaluation metrics.
4. The author studied and evaluated several auto-driving algorithms.

**Weaknesses:**

1. Could the users flexibly add new vehicle models to the platform? If yes, how?
2. Could the users add additional testing scenarios?
3. As for the benchmarking results, I suggest the author offer more analysis on these results and provide more insights into the different auto-driving algorithms.
4. Minor: I suggest the author include and add more adversarial attack methods in their platform, for example [1,2].

References

[1] Dual Attention Suppression Attack: Generate Adversarial Camouflage in Physical World. CVPR 2021.
[2] CAMOU: Learning physical vehicle camouflages to adversarially attack detectors in the wild. ICLR 2019.

**Additional Feedback:**

Please see the questions in the [Weakness].

**Documentation:**

Yes.

**Ethics:**

N/A.

**Relation To Prior Work:**

Yes.

**Summary And Contributions:**

This paper proposes SafeBench, the first platform as far as I know for the safety evaluation of auto-driving systems. In this platform, the author introduced several scenarios, auto-driving algorithms, and evaluation metrics. Based on the platform, the author conducted extensive experiments and offered some preliminary insights.

---

> ### Author Response · Authors · 2022-08-17
> **Thanks for your valuable comments**
>
> We thank the reviewer for recognizing our work novel and comprehensive and really appreciate the reviewer's suggestions to help improve the quality of our work. We answered the questions below and improved our paper following the suggestions.
>
> > **Q1.** Could the users flexibly add new vehicle models to the platform? If yes, how?
>
> Yes, users can follow the official tutorial from CARLA and our code repository instructions to add new vehicle models. We do not restrict the vehicle model used in our platform and more evaluations on different vehicle models are encouraged to comprehensively study the overall performance of AD algorithms. We also provide detailed instructions in our documentation.
>
> > **Q2.** Could the users add additional testing scenarios?
>
> Yes, users can add customized testing scenarios to the platform by simply providing the scenario configuration file. The Evaluation Node will automatically load these scenarios when testing. We also provide detailed instructions in our documentation.
>
> > **Q3.** As for the benchmarking results, I suggest the author offer more analysis on these results and provide more insights into the different auto-driving algorithms.
>
> Thanks for the insightful suggestions. In terms of the results of different RL algorithms, SAC shows the best benign performance while PPO shows the best safety-critical overall score. One of the possible reasons is that the training process of SAC and PPO is known to be more stable than the other two algorithms [1, 2]. Besides, DDPG and TD3 are sensitive to hyper-parameters. Simply changing the hyper-parameters will lead to a big performance drop. We will follow your suggestion and add more analysis on the results and different AD algorithms in the revision.
>
> > **Q4.** I suggest the author include and add more adversarial attack methods in their platform.
>
> We have incorporated several physical adversarial attack methods in revision Section 4.4. Specifically, we include point attack [3], scene attack [4], and pose attack to generate adversarial point cloud data. We also apply semantic transformations [5] to perturb multi-modal data. We will constantly add more testing scenarios and attacks to improve the comprehensiveness and reliability of SafeBench.
>
> —
>
> [1] Haarnoja, Tuomas, et al. "Soft actor-critic: Off-policy maximum entropy deep reinforcement learning with a stochastic actor." International conference on machine learning. PMLR, 2018.
>
> [2] Haarnoja, Tuomas, et al. "Soft actor-critic algorithms and applications." arXiv preprint arXiv:1812.05905 (2018).
>
> [3] Xiang, Chong, Charles R. Qi, and Bo Li. "Generating 3d adversarial point clouds." Proceedings of the IEEE/CVF Conference on Computer Vision and Pattern Recognition. 2019.
>
> [4] Ding, Wenhao, et al. "Semantically controllable scene generation with guidance of explicit knowledge." arXiv preprint arXiv:2106.04066 (2021).
>
> [5] Li, Linyi, et al. "Tss: Transformation-specific smoothing for robustness certification." Proceedings of the 2021 ACM SIGSAC Conference on Computer and Communications Security. 2021.

---

> ### Author Response · Authors · 2022-08-24
> **Post rebuttal discussion**
>
> We sincerely thank the reviewer for your previous insightful questions and suggestions, and we have tried our best to add additional evaluations/features to our platform as well as answer the questions. Please let us know if you have further questions or comments. We really look forward to your feedback to further improve our work. Thank you!

---

### Official Review · Reviewer_ijxU · 2022-07-26
**A useful simulator benchmark, but an unfocused paper**

**Rating:** 8
**Confidence:** 4
**Clarity:** The paper is clear and easy to read.

**Strengths:**

The benchmark software seems very useful for the autonomous vehicle research community and I can see the benchmark gaining a lot of traction. The software seems to be well written and the source code looks reasonable clean and well documented. It is also licensed under the MIT license and available via github.
The comparison of different algorithms is interesting and the drop in performance in the adverse scenarios invites future investigations.


**Weaknesses:**

My main problem with the paper is that is to me not clear what exactly it wants to be. The simulator is introduced, but I miss a detailed discussion of design decisions and an evaluation of the simulator itself.  For example, I miss a rational why the chosen scenarios are useful or better than some other scenarios. Why does it make sense to only keep scenarios that cause collision with at least two algorithms? Similarly, I miss an evaluation of the metric. Why is the metric good? How does it compare to other metrics?
Another point, I feel that should be elaborated is if simulator results are actually useful in real world system. So is an algorithm tested on this simulation better in a real world scenario? However, I am aware that such an evaluation is probably very complex and time consuming.

This is compounded by the excessive use of the appendix. It is ok to push some parts or background into the appendix, but at times the paper reads like a table of contents and most paragraphs seem to reference the appendix somehow.

It feels like the authors tried to do cram two papers into one; I am not sure if two papers are a better solution, but perhaps a longer journal paper would be an option? Another option might be to move the tables into the appendix and use the additional space to evaluate the simulator (and not the results of algorithms using the simulator!) and provide some more details on the design rational.

**Additional Feedback:**

line 127: a space after 'Figure 1'
line 128: references to docker and ros; also the acronym is never introduced

**Correctness:**

The simulator seems to be working and the source code is reasonably clean. The documentation seems complete and helpful.

The test results of the AD algorithms seem correct, however I do wish the authors would have provided a better rational for their metric.

**Documentation:**

The source code is available and released under an open source license (MIT)

**Ethics:**

There are no direct ethical problems with the simulator and the code, as all of the data is synthetic. However, the authors should consider how their simulator treats/introduces minorities. For example, it would be possible to include persons of color as pedestrians, or model disabled people like wheelchair users, which are from time to time mis classified by algorithms.

**Relation To Prior Work:**

Prior work is referenced. I just miss references to docker and ROS on line 128.

**Summary And Contributions:**

The authors propose a simulator environment that can be used to evaluate algorithms for autonomous driving according to 10 metrics.
This benchmark addresses the problem of testing AD algorithm in a standardized manner and seems to be very promising and useful for future research.

---

> ### Author Response · Authors · 2022-08-17
> **Thanks for your valuable comments (1/2)**
>
> We thank the reviewer for recognizing our work useful and inspiring and really appreciate the reviewer's suggestions to help improve the quality of our work. We answered the questions below and improved our paper following the suggestions.
>
> > **Q1.** The purpose of the paper is not clear. The simulator is introduced, but a detailed discussion of design decisions and an evaluation of the simulator itself are missing.
>
> We want to emphasize that we primarily focus on designing and developing a unified robustness and safety evaluation platform for AD algorithms in this paper. We build our platform based on CARLA, one of the most widely used simulators in AD. We have provided the platform design in Section 3, and we have added more details about our platform development in Appendix A.2 in our revision following the suggestion. We also add a qualitative table in revision Section 2 to summarize comparisons between our platform and existing evaluation platforms.
>
> > **Q2.** Why the chosen scenarios are useful or better than some other scenarios?
>
> We follow the NHTSA scenario ranking and the Carla Challenge to choose **safety-critical traffic scenarios** that have the highest rates of occurrence in the real world, which constitute the majority of the real-world scenarios. In addition, SafeBench is highly flexible, and users can also add customized scenarios with little additional effort. We will add the discussion in our revision.
>
> > **Q3.** Why does it make sense to only keep scenarios that cause collisions with at least two algorithms?
>
> As our goal is to select the possibly most safety-critical scenarios for efficient and effective AD testing, here we keep scenarios that can consistently cause accidents/collisions for more (at least two) algorithms. In general, testing scenarios that make more algorithms fail are riskier and more generalizable, which can potentially cause collisions for more algorithms during testing, making it more worthwhile testing. We will make this rationale clear in our revision.
>
> > **Q4.** An evaluation of the metric is missing. Why is the metric good? How does it compare to other metrics?
>
> We follow several previous works to design our comprehensive evaluation metrics.
> In particular, the *etiquette level* metrics are based on driving etiquette analysis [1,2] which encourages AD algorithms to behave like human-driven vehicles. The *safety level* metrics and *functionality level* metrics are based on CARLA Challenge [3] and MetaDrive [4] which also evaluate the safety and reliability of AD algorithms.
>
> We note that one evaluation metric is more important than the other under different scenarios, and here we aim to provide comprehensive evaluation metrics and analysis for different potential downstream usages. We will make this clear in our revision. This also makes SafeBench more comprehensive than other single evaluation tools.
>
>
> > **Q5.** Are simulator results actually useful in a real-world system? Is an algorithm tested on this simulation better than testing in a real-world scenario?
>
> Thanks for the insightful questions. Simulation is an important step before the expensive real-world testing and deployment, helping researchers and developers to identify critical flaws and vulnerabilities in their AD systems before testing in the real world, which may also cause more severe consequences; thus simulation is not replaceable for real-world tests. It is also generally acknowledged that a simulation is a useful tool for evaluating AD algorithms in practice [5, 6]. That is to say, both simulation and real-world tests are important considering their complementary properties.
>
> SafeBench follows the standard practice to build a unified and flexible evaluation platform that further supports the training of AD algorithms based on real-world data, improving the connection to the real world. We will add such discussion about the importance of simulation in the revision.
>
> > **Q6.** Excessive use of the appendix.
>
> We agree that the appendix should serve as supplementary material containing some relative information.  We have followed your suggestions and made the main paper more independent. Since we are allowed to add one page to the main context, we move some important information from the appendix to the main context.
>
> > **Q7.** The focus of the paper is unclear. It’s better to move the AD evaluation results to the appendix and add more details on the simulator and its design rationale.
>
> We agree that more details on the design of the evaluation platform should be helpful. We will follow your suggestions to include more design details and balance between different sections in the revision and move some details about the AD algorithms and tables into the appendix.

---

> > ### Author Response · Authors · 2022-08-17
> > **Thanks for your valuable comments (2/2)**
> >
> > > **Q8.** The authors should consider how their simulator treats/introduces minorities.
> >
> > Thanks for the valuable suggestion. As our evaluation platform is built based on CARLA, we can flexibly include pedestrians of different races and different conditions (e.g., wheelchair users) in our scenarios utilizing the pedestrian blueprints in CARLA. We add a diverse set of pedestrians with a variety of body shapes and skin colors in revision Section 4.4. We will add more features and keep maintaining our platform.
> >
> > —
> >
> > [1] Peng, Huei. Driving Etiquette. University of Michigan, Ann Arbor, Transportation Research Institute, 2020.
> >
> > [2] Huang, Xianan, Songan Zhang, and Huei Peng. "Developing robot driver etiquette based on naturalistic human driving behavior." IEEE transactions on intelligent transportation systems 21.4 (2019): 1393-1403.
> >
> > [3] CARLA Challenge, https://carlachallenge.org/.
> >
> > [4] Li, Quanyi, et al. "Metadrive: Composing diverse driving scenarios for generalizable reinforcement learning." IEEE Transactions on Pattern Analysis & Machine Intelligence 01 (2022).
> >
> > [5] ​​Amini, Alexander, et al. "Vista 2.0: An open, data-driven simulator for multimodal sensing and policy learning for autonomous vehicles." 2022 International Conference on Robotics and Automation (ICRA). IEEE, 2022.
> >
> > [6] Thorn, Eric, et al. A framework for automated driving system testable cases and scenarios. No. DOT HS 812 623. United States. Department of Transportation. National Highway Traffic Safety Administration, 2018.

---

> > > ### Comment · Reviewer_ijxU · 2022-08-17
> > > **Thank you for your revisions - change my score to clear accept**
> > >
> > > Thank you for your work, the detailed answers and the changes.
> > >
> > > I changed my reviewer score to 'clear accept'.

---

> > > > ### Author Response · Authors · 2022-08-24
> > > > **Thank you for your response**
> > > >
> > > > Thank you for your response! We really look forward to further improving our work. Please let us know if you have any other feedback or suggestions. Thank you!

---

### Official Review · Reviewer_NM99 · 2022-07-28
**Great Idea to increase AV Safety**

**Rating:** 8
**Confidence:** 3
**Correctness:** Yes, NA, yes.
**Clarity:** Yes, the paper is very clear.

**Strengths:**

The first platform of its kind as far as I know which is a huge contribution, the paper is very clear and well written, this paper has use-cases ranging from other researchers to actual AV companies, this platform could improve safety of AV's.

**Weaknesses:**

Central focus on just safety evaluation of AD makes the platform weaker (also labeling just the performance measures of safebench as true safety is misleading), traffic scenarios are not the only causes of unsafe driving, limited amount of AD algorithms tested, coming up with a single safety evaluation metric can be detrimental to learning/evaluation.

**Additional Feedback:**

I am curious as to how a person ports a model into the evaluation platform,

**Documentation:**

N/A, Yes.

**Ethics:**

Using a platform like SafeBench warrants ethical concern, as it could potentially be used to falsely deem autonomous vehicles as safe on roads when they are not in fact safe. It is important that if SafeBench were to become a standard for AV's that it would have much more comprehensive safety measures as well as approval from safety organizations.

**Relation To Prior Work:**

Yes, the related works section is small, but as this is a novel platform, there are limited prior works.

**Summary And Contributions:**

The authors of this paper propose a platform, SafeBench, for the safety evaluation of AV's. They claim this platform is the first of its kind, testing 8 different safety critical scenarios. Additionally, different AV algorithms are benchmarked on this simulation platform based on over 2000 safety scenarios. Results are generally promising, tested on simple RL algorithms, demonstrating tradeoffs between different AV metrics such as speed and running red lights.

---

> ### Author Response · Authors · 2022-08-17
> **Thanks for your valuable comments (1/2)**
>
> We thank the reviewer for recognizing our work novel and useful and really appreciate the reviewer's suggestions to help improve the quality of our work. We answered the questions as below and improved our paper following the suggestions.
>
> > **Q1.** It is insufficient to only focus on the safety evaluation of AD and one safety evaluation metric.
>
> We agree that comprehensive evaluation for AD is very important. In this work, we mainly aim to take the first step to provide a unified platform and benchmark for the *safety* evaluation of AD, since safety is currently one of the most concerning aspects of AD. Other evaluation goals would be interesting future work, which can be developed based on the proposed SafeBench platform as well.
> In addition, to fully assess the performance of AD algorithms from all aspects, we do take into account additional levels of evaluation metrics, such as **functionality level** and **etiquette level**. By calculating a weighted sum of metrics at all levels, we’re able to consider all the aspects and have a better understanding of the AD algorithm being tested.
>
> We will make such discussion clear in our revision.
>
> > **Q2.** It’s misleading to treat the performance measures of SafeBench as true safety. SafeBench could potentially be used to falsely deem autonomous vehicles as safe on roads when they are not in fact safe. It is important that if SafeBench became a standard for AVs, it would have much more comprehensive safety measures and approval from safety organizations.
>
> Thanks for the insightful comments. We agree that simulation in SafeBench cannot exactly reflect real-world conditions. Closing the sim-to-real gap has long been a challenge for existing simulators. Nevertheless, simulation is still popular in both research and industry and it is generally acknowledged that simulation can be a useful tool for evaluating AD systems, given its nice efficiency and controllability [1, 2]; and simulation is an important step before the expensive real-world testing and deployment, helping researchers and developers to identify critical flaws and vulnerabilities in their AD systems before the testing in the real world, which may cause more severe consequences. We follow the standard practice to build our unified evaluation platform based on CARLA, one of the most recognized simulators. Besides, with the flexible design of SafeBench, we can train AD algorithms based on real-world data, to better strengthen the connection to the real world. We will add more discussions following the suggestions in the revision.
>
> > **Q3.** Traffic scenarios are not the only causes of unsafe driving.
>
> Thanks for the comment. We agree that safety-critical traffic scenarios are only parts of the causes of unsafe driving. However, it has been shown that unsafe traffic scenarios are typically one of the main causes of unsafe driving [3]. Thus, in this paper, we primarily concentrate on safety-critical traffic scenario generation for AD testing and we will also make this scope clear.
> In addition, other causes can be incorporated into SafeBench as well. For example, we have incorporated several physical adversarial attack methods in revision Section 4.4 following the suggestion. Specifically, we include point attack [4], scene attack [5], and pose attack to generate adversarial point cloud data. We also apply semantic transformations [6] to perturb multi-modal data. We will constantly add more testing scenarios and attacks to improve the comprehensiveness and reliability of SafeBench.
>
> > **Q4.** There is a limited amount of AD algorithms tested.
>
> Here we mainly aim to generate diverse safety-critical scenarios to evaluate AD algorithms, thus in this initial release, we only evaluated a few widely used AD algorithms with a special focus on RL-based algorithms. We explore the different settings of these algorithms and draw some interesting conclusions. Because of our modular design, more complex AD algorithms can be easily tested in our framework. We will make it clear how to flexibly test other AD algorithms and keep integrating more AD algorithms on our platform.

---

> > ### Author Response · Authors · 2022-08-17
> > **Thanks for your valuable comments (2/2)**
> >
> > > **Q5.** How a person ports a model into the evaluation platform?
> >
> > With our modular design, users only need to provide a customized Agent Node with several basic APIs such as perception and planning. In addition, Agent Node is also made up of several modules so that users are able to flexibly replace any module including the end-to-end RL model. We have also provided detailed instructions in our documentation.
> >
> > —
> >
> > [1] ​​Amini, Alexander, et al. "Vista 2.0: An open, data-driven simulator for multimodal sensing and policy learning for autonomous vehicles." 2022 International Conference on Robotics and Automation (ICRA). IEEE, 2022.
> >
> > [2] Thorn, Eric, et al. A framework for automated driving system testable cases and scenarios. No. DOT HS 812 623. United States. Department of Transportation. National Highway Traffic Safety Administration, 2018.
> >
> > [3] Najm, Wassim G., John D. Smith, and Mikio Yanagisawa. Pre-crash scenario typology for crash avoidance research. No. DOT-VNTSC-NHTSA-06-02. United States. National Highway Traffic Safety Administration, 2007.
> >
> > [4] Xiang, Chong, Charles R. Qi, and Bo Li. "Generating 3d adversarial point clouds." Proceedings of the IEEE/CVF Conference on Computer Vision and Pattern Recognition. 2019.
> >
> > [5] Ding, Wenhao, et al. "Semantically controllable scene generation with guidance of explicit knowledge." arXiv preprint arXiv:2106.04066 (2021).
> >
> > [6] Li, Linyi, et al. "Tss: Transformation-specific smoothing for robustness certification." Proceedings of the 2021 ACM SIGSAC Conference on Computer and Communications Security. 2021.

---

> ### Author Response · Authors · 2022-08-24
> **Post rebuttal discussion**
>
> We sincerely thank the reviewer for your previous insightful questions and suggestions, and we have tried our best to add additional evaluations/features to our platform as well as answer the questions. Please let us know if you have further questions or comments. We really look forward to your feedback to further improve our work. Thank you!

---

### Review · Ethics_Reviewer_HuM4 · 2022-08-24

**Recommendation:** 1

**Ethics Review:**

While there are no flagrant ethical concerns for this paper, I would encourage the authors to include a limitations section to this benchmark to indicate it can not be assumed to be a comprehensive measure of all safety considerations for AVs. The responses the authors gave to NM99 are a step in the right direction, and this conversation can be better integrated in the conclusion section of the paper.

---

> ### Author Response · Authors · 2022-08-25
> **Thank you for your valuable comments**
>
> We thank the reviewer for the insightful suggestions. We follow your suggestions and add a *Limitation* section to section 5  in our revision. All the revisions are highlighted in blue. Please let us know if you have further questions or comments. Thank you!

---

### Author Response · Authors · 2022-08-17
**General Response**

We thank all the reviewers for their time and valuable suggestions. Following the suggestions from the reviewers, we have conducted additional experiments, corrected some typos, and made the illustration clearer in our revision. We will list our major updates below.

1. We added a new feature named **Robustness Toolkit** in Section 4.4, which applies physical semantic adversarial attacks to test different AD algorithms.
1. We added a comprehensive set of pedestrians with a variety of body shapes, skin colors, and rotation angles in Section 4.4.
1. We added a diverse set of vehicles with different colors, vehicle types, and rotation angles in Section 4.4.
1. We added 4 point cloud segmentation models and 2 multi-modal 3D object detection models for the perception of AD algorithms.
1. We added the experimental results of testing against point cloud segmentation attack modules in Table 5.
1. We added the experimental results of testing against attacks on multi-modal 3D object detection modules in Table 6.
1. We added visualization results of the robustness toolkit in Appendix A.10.
1. We added a systematic comparison between SafeBench and existing benchmark platforms in Section 2 and Table 1.
1. We included more design details of SafeBench in Appendix A.2.
1. We added an overall score for each level of evaluation metrics in Table 14.
1. We provided more computation complexity analysis of scenario generation algorithms and RL-based methods in Appendix A.11.
1. We clarified the source of the BEV image in Section 4.1.
1. We added the references and explanation for our evaluation metrics in Section 3.4.
1. We added more references for Docker and ROS on Line 128.
1. We added suggested related works and corresponding discussions in Section 2.

All of our revisions are updated in OpenReview and highlighted in blue. We look forward to more discussions. Thank you!

---

### Meta-Review · Area_Chair_ihkf · 2022-09-12

**Recommendation:** Accept
**Confidence:** 5

**Metareview:**

The paper addresses a much-needed benchmark for the safety of autonomous driving. All reviewers recognize the significance of the paper's contributions. Most of the questions have been addressed satisfactorily by the rebuttal. Overall, all reviewers are positive about the paper and recommend acceptance.

---

### Decision · Program_Chairs · 2022-09-16

Accept